

# Effects of mixing two legume species at seedling stage under different environmental conditions

Heba Elsalahy[1,2], Sonoko Bellingrath-Kimura[1,3], Timo Kautz[1] and Thomas Döring[4]

[1] Albrecht Daniel Thaer-Institute of Agricultural and Horticultural Sciences—Crop Science, Humboldt Universität Berlin, Berlin, Germany
[2] Faculty of Science, Botany and Microbiology Department, Assiut University, Assiut, Egypt
[3] Research Area "Land Use and Governance", Leibniz Centre for Agricultural Landscape Research (ZALF), Humboldt Universität Berlin, Müncheberg, Germany
[4] Agroecology and Organic Farming Group, Rheinische Friedrich-Wilhelms Universität Bonn, Bonn, Germany

Corresponding author
Heba Elsalahy,
Heba.elsalahy@agrar.hu-berlin.de

## ABSTRACT

While intercropping is known to have positive effects on crop productivity, it is unclear whether the effects of mixing species start at the early plant stage, that is, during germination. We tested whether the germination of two legume species, alsike clover and black medic, characterized by a contrasting response to water availability and temperature is affected by mixing. We set up four experiments in each of which we compared a 1:1 mixture against the two monocultures, and combined this with various other experimental factors. These additional factors were (i) varied seed densities (50%, 100% and 150% of a reference density) in two field trials in 2016 and 2017, (ii) varied seed densities (high and low) and water availability (six levels, between 25% and 100% of water holding capacity (WHC)) in a greenhouse pot trial, (iii) varied seed spacing in a climate chamber, and (iv) varied temperatures (12 °C, 20 °C and 28 °C) and water availability (four levels between 25% and 100% of WHC) in a climate chamber. Across all experiments, the absolute mixture effects (AME) on germination ranged between −9% and +11%, with a median of +1.3%. Within experiments, significant mixture effects were observed, but the direction of these effects was inconsistent. In the field, AME on germination was significantly negative at some of the tested seed densities. A positive AME was observed in the climate chamber at 12 °C, and the mean AME decreased with increasing temperature. Higher density was associated with decreased germination in the field, indicating negative interaction through competition or allelopathy, among seedlings. Our findings indicate that interaction among seeds in species mixtures may be ongoing during germination, but that the direction of the mixture effect is affected by complex interactions with abiotic and biotic factors.

## INTRODUCTION

Diverse cover crop species are used in agriculture to provide various ecosystem services, such as increasing soil fertility for the next crop, nutrient recycling, suppressing weeds, and protecting soils against erosion (*Döring et al., 2013*; *Elsalahy et al., 2019*). Using these species in mixtures can increase the benefits that are conferred by the cover crops (*Döring et al., 2013*; *Stagnari et al., 2017*; *Storkey et al., 2015*). Further, many cover crops are great forage crops, and using them in binary mixtures demonstrated high yield and nutritive value of the selected species (*Aponte, Samarappuli & Berti, 2019*; *Bélanger et al., 2017*; *Cherney et al., 2020*; *Waldron et al., 2020*). Thereby, cover crop mixtures may constitute a promising future for sustainable agricultural systems and forage production.

One challenge is to successfully establish cover crop stands in a competitive environment, for example, when the temperature is too low or too high and the amount of precipitation is too low for optimum growth conditions. In particular depending on sowing date, environmental conditions may vary strongly during early plant development. Studies on cover crop mixtures demonstrated that mixing species with asynchronous growth, that is, the dominance of different species at different points in time may provide positive mixture effects on the expected services in unpredictable conditions (*Elsalahy et al., 2020*; *Mori, Furukawa & Sasaki, 2013*). This approach may reduce species competitiveness under conditions of limited growth and allow for complementary crop growth (*Elsalahy et al., 2020*).

However, in cropping systems, it has not been demonstrated yet whether there are mixture effects on seed germination and seedling emergence, that is, at the very beginning of plant interactions. This is important since it is known that the ability of a species to germinate faster than its neighbours may affect species performance and dominance in the mixture (*Goldberg & Landa, 1991*; *Fenesi et al., 2020*; *Tielbörger & Prasse, 2009*; *Verdú & Traveset, 2005*). A simple example of making use of mixture effects in the context of germination is risk reduction; plants vary in temperature and water requirements for germination, hence choosing species with contrasting germination abilities and requirements would decrease the risk of non-emergence in unpredictable environments (*Tribouillois et al., 2016*). Understanding the different aspects of seed germination in mixtures under different environmental conditions may thus help to secure successful crop establishment and to promote positive mixture effects.

Temperature and water availability are particularly important factors that regulate germination (*Hatfield & Prueger, 2015*; *Luo et al., 2018*; *Tribouillois et al., 2016*). In addition to the environmental quality, the biotic environment, in particular, the density and identity of neighboring seeds or seedlings may affect the behavior of seed germination in the mixture (*Goldberg & Landa, 1991*; *Leverett, Schieder & Donohue, 2018*; *Tielbörger & Prasse, 2009*). Plants use a number of cues allowing them to assess the best time for germination in intra- and interspecific competition, for example, the moisture content, proper temperature, or the density of the neighbour seeds (*Bergelson & Perry, 1989*; *Tielbörger & Prasse, 2009*). An important component of this plasticity is accelerated

seedling emergence in competitive neighborhoods (*Sales, Pérez-García & Silveira, 2013*; *Tielbörger & Prasse, 2009*; *Verdú & Traveset, 2005*). The potential mechanisms of seed germination plastically responding to neighbors may be attributed to the fact that germinating seeds may emit volatile organic compounds (VOCs) based on the growing conditions (*Fincheira, Parada & Quiroz, 2017*; *Motsa et al., 2017*). Studies showed that temperature level affects production rates of VOCs and may change the volatile composition via affecting enzyme activity that degrades the storage reserves in the seeds; hence affects breaking seed dormancy (*Effah, Holopainen & McCormick, 2019*; *Motsa et al., 2017*). It is therefore likely that environmental conditions may have an effect on the interactions among seeds, and thereby on potential mixture effects.

We, therefore, tested whether or not the germination of the only-legume mixture is affected by the identity of neighboring seeds or seedlings under different temperature and moisture conditions. In particular, we studied the mixture effect on germination for two perennial forage legume species that have different responses to water and temperature requirements. We chose alsike clover (AC; *Trifolium hybridum* L.) and black medic (BM; *Medicago lupulina* L.) and a 1:1 mixture of the two species as previous studies showed a positive mixture effect of these two species at this proportion regarding productivity, weed suppression, and resilience to drought (*Elsalahy et al., 2019*; *Elsalahy et al., 2020*). BM is a fast-growing perennial well adapted to warm and dry areas (*Döring et al., 2013*; *Elsalahy et al., 2019*; *FAO, 2020*; *Komainda et al., 2019*) with a base temperature of 0.6 °C and optimal germination temperature of 26 °C (*Tribouillois et al., 2016*). Nonetheless, it was also reported for BM that the optimal range of temperatures lies between 10 °C and 20 °C (*Sharpe & Boyd, 2019*). Conversely, AC is a more slowly growing and comparatively drought-sensitive perennial (*Chapman, Dodds & Keoghan, 1990*), with less tolerance to drought or high temperature (*Sheaffer et al., 2003*), and best adapted to cool and wet areas (*Döring et al., 2013*; *Nation, 1989*). The minimal germination temperature of AC has been given as 2 °C (*Kahnt, 2008*) or 5 °C (*Hartmann & Lunenberg, 2013*). In addition, both of the species are classified as cool-season legumes, namely, they are characterized by decreasing germination rate at temperatures greater than the optimal temperature range (*Butler et al., 2014*).

We hypothesized that germination would be higher in the mixture than in the respective monocultures. This was mainly based on the assumption of relaxed competitive pressure on germinating plants. However, in a complex system, for example, in the field, the competition between the germinated seeds and germinated weeds, at the early stage, may interact with environmental conditions, and drive mixture effect in a specific direction. To test our hypothesis, we used a set of experiments to observe the germination of the monocultures and the mixture at different seed densities, drought intensities, temperature levels, and seed spacing. Field trials over two years were used to represent the most complex environment, whereas pot and laboratory trials were designed to limit specific sources of variation. This range of different trials allowed us to test our central hypothesis under different environmental conditions, and varied environmental complexity, which ranged from low, as in the climate chambers, to high, as in the field.

## MATERIALS AND METHODS

### General set-up

Two field trials, one pot trial, and two laboratory trials were performed to study potential mixture effects on germination of two legume species AC (cv. Dawn) and BM (cv. Ekola), and an equiproportional (1:1) substitutive mixture of the two species. The seeds of AC and BM were bought from Deutsche Saatveredelung AG (DSV) and Camena Samen, Germany, respectively. For all trials, seeds were left unsterilized before sowing to mimic common agricultural practice. In the field, the seed placement of the two species, in the mixture, was spatially random based on the distribution of the seeds by the plot drill in the rows of each plot. In contrast, in the pot and laboratory trials, the seeds of the two species, in the mixture, were precisely spatially alternated to ensure maximal interspecific interaction.

### Field trials

To measure potential mixture effects of germination and seedling emergence under field conditions, two field trials were conducted in 2016 (field FU9) and 2017 (field S5), respectively. Both trials were conducted at the experimental field station of the Humboldt University of Berlin in Dahlem (52° 28′ N, 13° 18′ E, 51 m asl). Soil fertility and soil texture varied strongly between fields, specifically sandy clay loam and sandy loam were found in field FU9 and field S5, respectively (*Elsalahy et al., 2019*). The crop sequence in both field trials was cereals—potato—winter wheat—legumes.

The experimental design in both years was a randomized complete block design with three replicates in 2016 and four replicates in 2017, as previously described in *Elsalahy et al., 2019*, with the factor diversity (called DIV) comprising three diversity treatments (two monocultures and one mixture) and three different sowing densities (called Den) representing 50%, 100% and 150% of typically recommended seed density (i.e., the reference density). The number of blocks increased to 4 in 2017 to increase the precision of the findings from the year 2016. Seeding rates were 7.6 and 17.6 kg ha$^{-1}$ of AC and BM, respectively, and their thousand-grain weight was 0.80 g for AC and 1.85 g for BM (*Elsalahy et al., 2019*). The absolute number of seeds in the monocultures at 150% seed density was approximately 1600 seed m$^{-2}$. The seeds were weighed and mixed thoroughly in case of the mixture and kept at room temperature (approximately 20 °C) for a few days before sowing. Sowing time was on 29th and 25th April in 2016 and 2017, respectively. Sowing depth was 0.5 cm and was adjusted precisely in the field by using a sowing machine (Wintersteiger AG, Ried im Innkreis, Austria). Plot size was 3 m × 9 m including plot margins. Each plot consisted of 20 rows with 13.5 cm spacing between rows.

Counting of emerged seedlings was performed in a number of selected 0.5 m long sections from the middle of the plots, that is, counting was only done within the central 7 m of the total 9 m plot length, and omitting the outer rows, thereby avoiding potential edge effects (Fig. S1). The total number of samples per each of the 9 variants (3 DIV × 3 Den) was $n = 24$ (eight sections × three blocks) in 2016 and $n = 48$ (12 sections × four blocks) in 2017. The time of counting was based on calculating the growing degree days

(GDD; °Cd) of BM by using a base temperature of 0.6 °C (*Tribouillois et al., 2016*) to compare plants in similar growth stages across experimental years. Depending on these calculations the plants were specifically counted at 390 °Cd (25 days after sowing; DAS) and 381 °Cd (28 DAS) in 2016 and 2017, respectively. The plants in the counting areas were manually separated into three fractions of AC, BM and all other species which were considered as weeds.

## Pot trial

The trial was started on 1st June 2016. The greenhouse conditions, the experimental design, and the irrigation strategy of the pot experiment were previously described in *Elsalahy et al., 2020*. Specifically, the greenhouse was protected on all sides with a wire mesh woven in a plain weave pattern (25.4 × 25.4 mm clear opening, 3.18 mm Ø wire). The size of the openings (spaces) of the wire mesh allowed insects, the wind, and temperature to be in the greenhouse as natural as in the field but prevented any animals such mice from entering. The roof of the greenhouse was covered with polycarbonate plastic panels (16 mm triple wall) that allowed 76% light transmission. The average daily temperature during the considered period for germination was 19.2 °C, ranging from 10.4 °C to 28.3 °C (*Agricultural Climatology of the Humboldt-University of Berlin, 2019*).

The experimental design was a three-factorial randomized complete block design in four replicates with the factor diversity (DIV; 3 diversity treatments), the density factor (called Den) with low and high seed densities, and the drought factor (called CD), with 6 levels of cumulative drought intensity (100%, 85%, 70%, 55%, 40% and 25% of water holding capacity (WHC)). Each block contained 18 pots and one control pot without plants was used to calculate the daily evaporated water. The soil was taken from the top-soil (upper ~15 cm) of a non-cropped bare field (field FU9) at the experimental station of the Humboldt University of Berlin in Dahlem. The soil was a sandy clay loam (*Brady & Weil, 2002*), had a pH of 6.3, organic matter content of 1.24%, N content of 0.13%, and nutrient contents per kg soil of 251 mg P, 90 mg K, 52 mg Mg, 1471 mg Ca, and 7354 mg $Fe^{3+}$. Soil WHC was measured in three replicates and calculated following (*Nguyen & Lehmann, 2009*).

One day before sowing, square pots (12 cm × 12 cm × 19 cm height) were filled with 3430 g of the soil. Then, on the day of sowing, 478 ml distilled were added to each pot to moisten the soil and to facilitate placing the seeds in fixed distances without silting up the soil surface. Seeds were sown at a depth of 0.5 cm. A handmade wooden seed stamp was used to adjust the space between sown seeds in a design of 3 × 4 (4 cm space; 12 seeds $pot^{-1}$) and 6 × 4 (2 cm space; 24 seeds $pot^{-1}$) at low and high seed densities, respectively. These densities were roughly equivalent to seed densities at 77.5% and 155% of the reference density in the field. Three days later, an equal amount of water was added to all treatments to compensate for evaporation and ensure optimum conditions for germination in all treatments by keeping WHC at ~90% before starting the different irrigation regimes.

In the second week after sowing, differentiated irrigation was initiated, creating six different levels. Precise irrigation was facilitated by using a dispenser (Rotilabo®-Dispenser

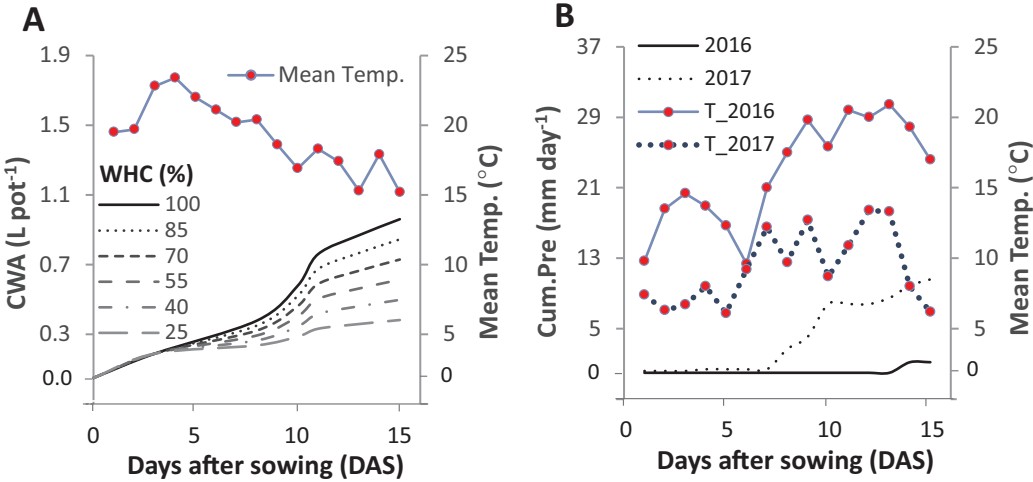

**Figure 1 Cumulative water applied per pot (CWA; A) in the pot trial and the cumulative precipitation (Cum.Pre; B) in two field trials after fifteen days of sowing.** The pot trial started on 1st June 2016 and included six intensities of cumulative drought: 100%, 85%, 70%, 55%, 40% and 25% water holding capacity (WHC). The two field experiments were started on the 29th and 25th of April in 2016 and 2017, respectively. The mean daily temperature (Mean Temp.) is represented in solid line for 2016 and in the dotted line for 2017.

20–100 ml, accuracy 1 ml, by Roth, Germany). The average weight of the control pots (without plants) was determined to represent the amount of evaporated water. This amount was then given to the 100% WHC level and reduced proportionally based on the level of drought intensities (Fig. 1A). Germinated seeds were counted fifteen days after sowing. Any emerged weed seedlings were removed daily from the pots.

## Laboratory trial 1

This trial tested the effects of two factors, species mixing (DIV) and seed spacing (called SPA) in a factorial set up. As in the other trials, the factor DIV contained three levels (AC, BM, and a 1:1 mixture of both species). The factor seed spacing had two levels, called "single" and "double" seeds. In the single seed treatment, single seeds were equally spaced with a 1 cm distance between seeds. In the double seed treatment, seeds were sown in pairs, that is, as closely together as possible (~1 mm distance); depending on the level of the mixture treatment these pairs of seeds either contained two seeds of the same species or two seeds of different species. Pairs were spaced 2 cm apart, so that the overall seed density was the same for the two spacing treatments. The trial was set up in a randomized block design with 12 replications.

Seeds were sown in shallow plastic trays (50 × 30 cm), by manually placing them onto a layer of 2 cm deep sieved standard growing substrate ("Terrasoil", DIN 18035 part 4 and RAL GZ 515/2, Cordel-Bau GmbH, Wallenborn, Germany), and covering them with 1 cm of the same substrate. This resulted in a total volume of 4.5 L substrate per tray. Each tray constituted one block; 18 seeds were sown per treatment per block, resulting in a total of 12 × 18 = 216 seeds per treatment. Directly after sowing, each tray was carefully irrigated with 350 ml tap water. Trays were then kept in a climate chamber with a constant

temperature at 20 °C and a light regime of 12/12 h. Every day each tray was watered with 275 ml. Germination was evaluated daily, but here we only present the final germination rate, as the percentage of developed viable seedlings 2 weeks after sowing.

### Laboratory trial 2

The experiment was conducted in growth chambers (RuMed Rubarth Apparate GmbH, Typ 3501, Germany) under precise control of temperature levels and dark conditions. The experimental design of the germination experiment was a three-factorial randomized complete block design in four replicates. The experimental factors comprised the factor diversity (DIV; three levels: AC, BM and the mixture, as above), the drought factor (called CD), with four levels of drought intensity (100%, 75%, 50% and 25% WHC), and temperature factor (Called $T$), with three levels of 12 °C, 20 °C and 28 °C. Thus, each block contained 36 treatments. The soil was collected from the same area used for the pot trial, sieved through a sieve 5 mm mesh size, dried and sterilized in the oven (Thermo Scientific Heraeus UT 6760, Germany) at 105 °C for 48 h.

Rectangular plastic boxes with a volume of 750 cm$^3$ (15 cm length × 10 cm width × 5 cm height) were filled with 100 g of the sterilized soil, which was moistened with the calculated WHC of the sterilized dry soil. The amount of water required to initiate the drought intensities was added once by using distilled water at 43.3, 32.5, 21.6 and 10.8 ml box$^{-1}$ for the levels 100%, 75%, 50% and 25% of WHC, respectively. The seed density was 24 seeds box$^{-1}$ which was equivalent to the 150% seed density in the field (see above). The seeds were placed on the surface of the moistened soil and placed in dark incubators under controlled temperatures. The distances between the seeds were fixed by using a handmade seed stamp of 100 microns thick transparent sheet (DATALINE transparency film, EU) with an area of 15 cm × 10 cm. The stamp was pierced in a design of 6 × 4 to have a 2 cm space between the seeds. The boxes were covered with their transparent plastic covers to prevent moisture loss and were then randomized within the shelves in the growth chamber.

The temperature levels were constant during the experimental period as studies on BM reported no significant difference in its germination percentage at constant vs. fluctuating temperatures (*Sharpe & Boyd, 2019*; *Tribouillois et al., 2016*). Based on calculating the growing degree days (GDD; °Cd) of BM of 0.6 °Cd (*Tribouillois et al., 2016*) for the 5 experimental days, the seeds were counted after accumulating 68, 185 and 349 °Cd at temperatures 12 °C, 20 °C and 28 °C, respectively.

The trial was replicated three times with renewing the sterilized soil and interchanging the temperature levels between the growth chambers to avoid potential incubator effects. Germination assessment was done by counting the germinated seeds once after 5 days of sowing. A seed was considered to have germinated when its radicle had been pushed out with a length of ≥ 0.2 cm.

### Calculation of indices

The effect of mixing seeds on germination was estimated at the different climate conditions by considering the absolute mixture effects (AME), that is, the difference between the

counted seeds in the mixture and the average counted seeds of the two monocultures (*Elsalahy et al., 2019*; *Elsalahy et al., 2020*) as in Eq. (1).

$$AME = y_{\text{mix}} - (y_{\text{AC}} + y_{\text{BM}})/2 \qquad (1)$$

In analogy to the frequently used Land Equivalent Ratio (LER) and partial LER that were calculated in *Elsalahy et al. (2020)*, the germination ratio (GR) and the partial germination ratios (pGR) of each species were calculated, in order to quantify the contribution and dominance of each species in the mixture in Eq. (2).

$$GR = PGR_{\text{AC}} + PGR_{\text{BM}} = \frac{g_{\text{AC\_mix}}}{g_{\text{AC\_mono}}} + \frac{g_{\text{BM\_mix}}}{g_{\text{BM\_mono}}} \qquad (2)$$

where $g_{\text{AC\_mono}}$ and $g_{\text{BM\_mono}}$ are the germinated seeds of species AC and BM in monoculture and $g_{\text{AC\_mix}}$ and $g_{\text{BM\_mix}}$ are the germinated seeds of each species in the mixture. A GR > 1 indicates that the mixture is more efficient in terms of germination than the monocultures. Partial GRs can be interpreted as a measure for the contribution of each species according to its density ratio in the mixture relative to the monoculture.

A nonlinear mixed-effects regression model with two parameters (Weibull function) was used to estimate the germination percentage of AC and BM in monocultures and mixtures at the different temperature levels as a function of water availability WHC using Eq. (3).

$$y = a*(1 - \exp(b*x)) \qquad (3)$$

where $y$ is germination percentage, $x$ is the WHC level, and $a$ and $b$ are estimated parameters, with $a$ being the upper asymptote, that is, representing the maximal germination percentage.

## Statistical analysis

The steps of the statistical analysis, the selected models, and using some specific tests for distinct response variables were previously described in *Elsalahy et al. (2020)*. Specifically, a generalized linear model (GLM) with binomial error distribution and a probit link function was used to analyze germination in all trials. In the field trial, pot trial, and laboratory trial 2, as a first step a full model was used with considering the main factors and all possible interactions as fixed effects. Subsequently, to simplify interpreting the results, further submodels were used by considering two main factors and the interaction between them. In these trials, block effect was removed from the model as it did not improve the model according to Akaike's information criterion (*Burnham, Anderson & Huyvaert, 2011*). In the full model and each sub-model, the normality of residuals was checked by using Shapiro test.

In the two field trials, the full model was constrained to DIV, Den, *Year* and all possible interactions. Then, to evaluate the germination of the plants in the monocultures and the mixture at different seed densities in response to different environmental conditions, the submodels included DIV, Den and DIV × Den. In the pot trial, the full model was

constrained to DIV, CD, Den and all possible interactions. Subsequently, to evaluate plant response to water availability at different seed densities, the submodels were limited to DIV, CD and DIV × CD. In laboratory trial 2, the full model was constrained to DIV, CD, *T* and all possible interactions. Subsequently, to evaluate plant response to water availability at different temperatures, the submodels constrained DIV, CD and DIV × CD. However, in laboratory trial 1, only a full model was used that constrained DIV, SPA and DIV × SPA.

In the laboratory trial 2, to plot the germination response curves at the different temperatures, an extended non-linear regression model (with lower limit at 0 and upper limit estimated at 100) was fitted as described in Eq. (3). As the variance was not homogenous in this case, the correlation = corAR1 function was added to the model using the *nlme* package (*Pinheiro et al., 2018*). Goodness-of-fit between predicted and observed data was assessed by means of the coefficient of determination ($R^2$).

The significance of AME was determined, after ANOVA, by Tukey's HSD test ($\alpha = 0.05$) by using the *Agricola* R- package (*De Mendiburu, 2019*). Specifically, in the field trials, AME was compared at different seed densities (Den) in each year, in the pot trial, at different drought intensities (CD), in laboratory trial 1, at different seed spacing (SPA), and in laboratory trial 2, at different temperatures (*T*). In all trials, the significance of GR being ≠1 was tested by using two-sided Welch's t-tests against 1. All statistics were performed using R (version 3.6.1) with R studio (version 1.1.463) (*R Core Team, 2019*).

# RESULTS

## Germination under field conditions

In the field trials, the germination percentage was significantly different between years. Splitting the two years showed that the effects of DIV, Den and DIV × Den were significant (Table 1). On average of all diversity treatments, the germination was 34.4% in 2016 in comparison with 54.4% in 2017. In 2016, increasing seed density reduced germination of AC, BM, and Mix by 13.2%, 26.7% and 39.4%, respectively (Fig. 2A), while in 2017, gradually increasing seed density reduced the germination of AC, BM, and the Mix by 20.1%, 18.0% and 33.6%, respectively. Thus, the germination rate was negatively affected by seed density in both years, and in both species and the mixture. In addition, a higher number of weed seedlings were found in 2017 than in 2016.

The absolute mixture effect was significantly different in both years. Specifically, in 2017, a stronger negative AME was observed than in 2016 (Fig. 2B). In 2016, only at 100% seed density, the AME was significantly negative, that is, the germination percentage in the mixture was significantly lower than the monocultures by 5%, while, in 2017, at 100% and 150%, germination was significantly reduced in the mixture by 5.5% and 9.4%, respectively (Fig. 2B). The field trial showed no significant deviation of GR from 1 in any of the years. In addition, both of the species germinated in the mixture according to the expected ratio, that is, almost equal to 0.5 (Figs. 2C and 2D) except in 2016, at 50% seed density, when BM was dominant.

**Table 1 Analysis of deviance on effects fitted to germination percentage of AC and BM and a 1:1 mixture of the two species in the field and pot trials.**

| Experiment | Model | DF | Deviance | Residual DF | Residual deviance | F-value | P-value |
|---|---|---|---|---|---|---|---|
| Two fields | NULL | | | 647 | 81590.0 | | |
| | Plant diversity (DIV) | 2 | 5842.2 | 645 | 75747.0 | 2921.1 | <0.0001*** |
| | Density (Den) | 2 | 7280.9 | 643 | 68467.0 | 3640.5 | <0.0001*** |
| | Year (Y) | 1 | 25498.2 | 642 | 42968.0 | 25498.2 | <0.0001*** |
| | DIV × Den | 4 | 787.4 | 638 | 42181.0 | 196.9 | <0.0001*** |
| | DIV × Y | 2 | 1741.1 | 636 | 40440.0 | 870.5 | <0.0001*** |
| | Den × Y | 2 | 706.3 | 634 | 39734.0 | 353.2 | <0.0001*** |
| | DIV × Den × Y | 4 | 305.3 | 630 | 39428.0 | 76.3 | <0.0001*** |
| Field_2016 | NULL | | | 323 | 8461.2 | | |
| | Plant diversity (DIV) | 2 | 190.9 | 321 | 8270.3 | 95.5 | <0.0001*** |
| | Density (Den) | 2 | 9.1 | 319 | 8261.1 | 4.6 | 0.01* |
| | DIV × Den | 4 | 72.6 | 315 | 8188.6 | 18.1 | 0.0*** |
| Field_2017 | NULL | | | 755 | 44131.0 | | |
| | Plant diversity (DIV) | 2 | 1280.9 | 753 | 42850.0 | 640.4 | <0.0001*** |
| | Density (Den) | 2 | 1568.2 | 751 | 41282.0 | 784.1 | <0.0001*** |
| | DIV × Den | 1 | 19231.4 | 750 | 22051.0 | 19231.4 | <0.0001*** |
| Pot | NULL | | | 142 | 217.7 | | |
| | Plant diversity (DIV) | 2 | 22.7 | 140 | 195.0 | 11.3 | 0.0*** |
| | Cumulative drought (CD) | 5 | 5.9 | 135 | 189.2 | 1.2 | 0.32 |
| | Density (Den) | 1 | 3.5 | 134 | 185.7 | 3.5 | 0.06 |
| | DIV × CD | 10 | 13.5 | 124 | 172.1 | 1.4 | 0.2 |
| | DIV × Den | 2 | 1.2 | 122 | 170.9 | 0.6 | 0.55 |
| | CD × Den | 5 | 2.1 | 117 | 168.8 | 0.4 | 0.82 |
| | DIV × CD × Den | 10 | 14.0 | 107 | 154.7 | 1.4 | 0.17 |

**Note:**
The analysis used a GLM with binomial error distribution and probit link function (significant codes: *** = $P < 0.001$ and * = $P < 0.05$).

## Germination under greenhouse conditions

In the pot trial, the DIV effect was significant at $p < 0.05$, and the density effect (Den) was almost significant at $p < 0.1$, while the drought effect (CD) did not affect the germination percentage of any of the monocultures and the mixtures (Table 1).
At both densities, the germination of $BM_{mono}$ was higher than $AC_{mono}$ with a difference of 5.2%, on average over all drought intensities (Figs. 3A and 3B). On average of all diversity treatments and drought intensities, increasing seed density reduced the germination by 4% (Fig. 3G).

The AME was > 0 and GR > 1 only at 70% WHC at high density, showing that the germination in this mixture was improved compared to the monocultures only at this level of water availability, but not at the other conditions (Figs. 3B–3E). On average, AME was positive at both densities and higher at high density but this difference was not significant (Fig. 3H). The change in PGRs of both species at the different drought

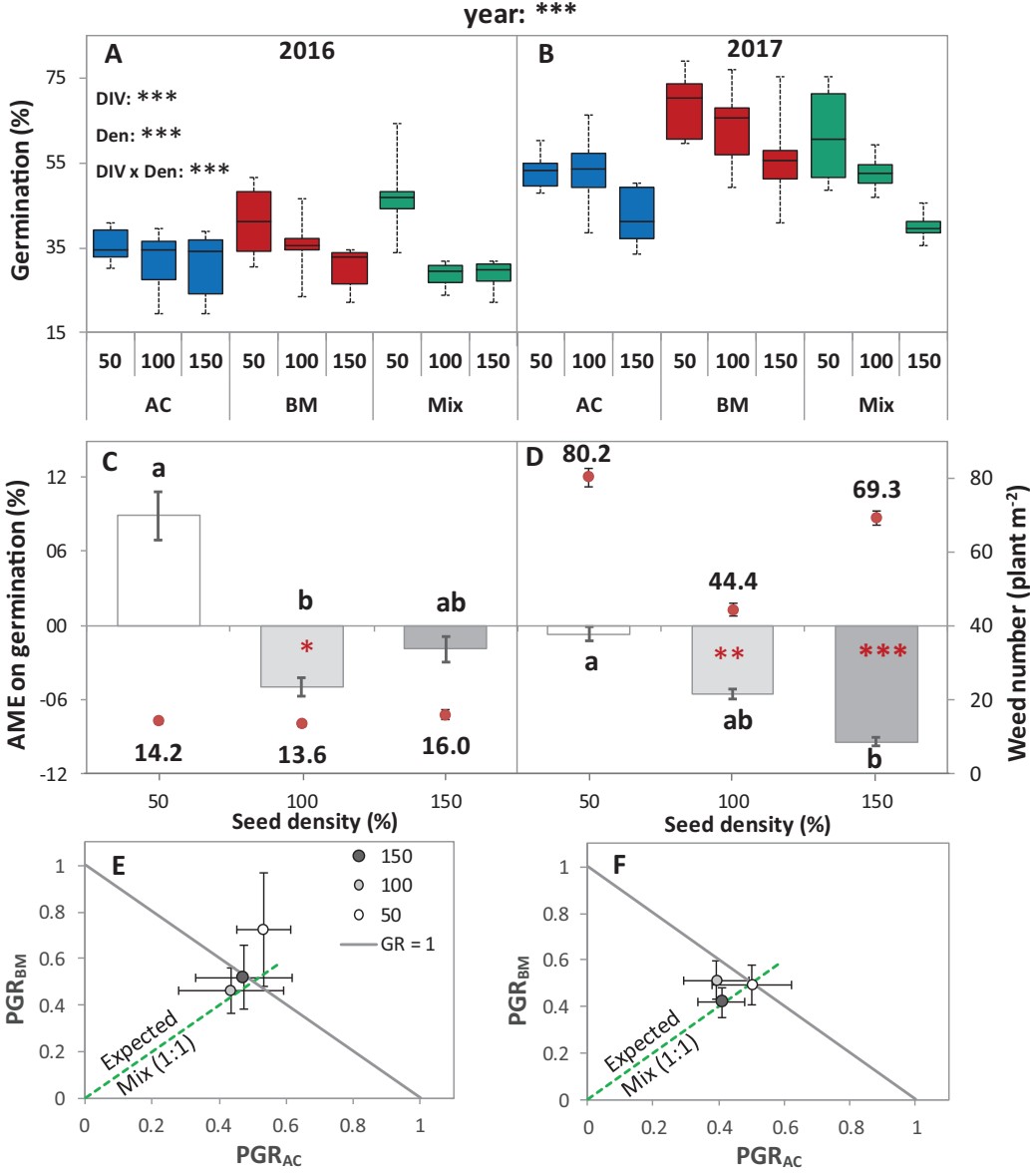

**Figure 2 Germination of alsike clover (AC) and black medic (BM) in monocultures, and a 1:1 mixture (Mix) of the two species in two field trials conducted in 2016 and 2017.** Germination percentage (%; A and B), absolute mixture effect (AME; C and D), and partial germination ratio of both species (PGR; E and F) are calculated for seeds sown at three seed densities (50%, 100% and 150% of recommended seed density). The boxes of the boxplot represent the median, and 25th and 75th percentiles, whereas whiskers represent the maximum and minimum values. The absolute mixture effect was calculated by Eq. (1); the dots above each column represent the number of germinated weeds m$^{-2}$. Negative values indicate a negative mixture effect on germination and vertical bars represent SE of $n = 24$ and 36 in 2016 and 2017, respectively. Different letters above bars indicate significant differences among differences of AME among the different seed densities. Black asterisks beside the main effects represent the analysis of deviance, while the red asterisks above the error bars represent a significant AME than 0 according to Welch's $t$-test; ***$P < 0.001$, **$P < 0.01$, *$P < 0.05$. The partial germination ratio was calculated by Eq. (2); the solid grey line corresponds to a GR = 1 and the broken green line represents the expected PGR for the mixture (0.5).

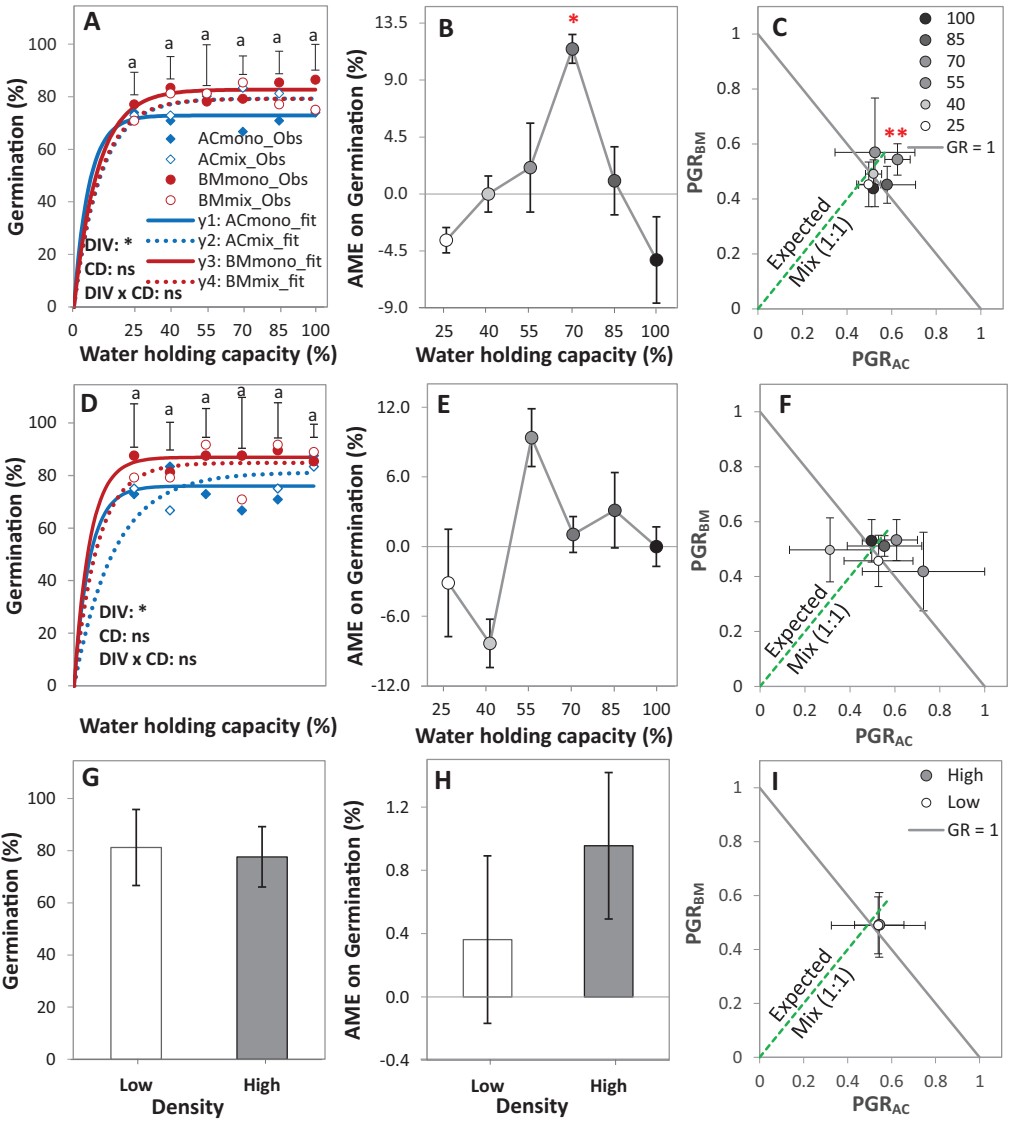

**Figure 3 Germination of alsike clover (AC) and black medic (BM) in monocultures, and a 1:1 mixture (Mix) of the two species in a pot trial.** Germination percentage (%; A, D and G), absolute mixture effect (AME; B, E and H), and partial germination ratio of both species (PGR; C, F and I) are calculated for seeds sown at two seed densities: high density of 24 seed pot-1 (A–C) and low seed density of 12 seed pot$^{-1}$ (D–F) under six intensities of cumulative drought (CD; 100%, 85%, 70%, 55%, 40% and 25% WHC). The observed germination is represented in symbols and the fitted curves (Eq. (3)) are in lines ($n = 4$); vertical bars represent Tukey's HSD ($p < 0.05$) among the monocultures (mono) and the mixtures (Mix) at a given WHC (A and D). Different letters above the vertical bars represent Tukey's HSD ($p < 0.05$) for the differences among the WHC (%). Asterisks indicate significant effects of the *DIV, CD* and *DIV × CD* based on the analysis of deviance for the generalized linear model; **$P < 0.01$, *$P < 0.05$, ns = not significant. The absolute mixture effect was calculated by Eq. (1); the drought intensities were visualized in black gradient color. Positive values indicate a positive mixture effect. The Partial germination ratio was calculated by Eq. (2); the solid grey lines and the broken green lines as in Fig. 2. Asterisks above some data points represent a significant increase in AME > 0 or GR > 1 according to Welch's *t*-test.

**Table 2 Analysis of deviance on effects fitted to germination percentage of AC and BM and a 1:1 mixture of the two species in different the two laboratory trials.**

| Trial | Model | DF | Deviance | Residual DF | Residual deviance | *F*-value | *P*-value |
|---|---|---|---|---|---|---|---|
| Laboratory 1 | NULL | | | 71 | 237.8 | | |
| | Plant diversity (DIV) | 2 | 7.2 | 69 | 230.6 | 3.6 | 0.03* |
| | Spacing (SPA) | 1 | 3.7 | 68 | 226.9 | 3.7 | 0.06 |
| | DIV × SPA | 2 | 4.4 | 55 | 96.1 | 2.2 | 0.11 |
| Laboratory 2 | NULL | | | 431 | 2602.8 | | |
| | Plant diversity (DIV) | 2 | 180.2 | 429 | 2422.6 | 90.1 | <0.0001*** |
| | Cumulative drought (CD) | 3 | 1016.6 | 424 | 1031.1 | 338.9 | <0.0001*** |
| | Temperature (T) | 2 | 374.9 | 427 | 2047.7 | 187.4 | <0.0001*** |
| | DIV × CD | 6 | 4.2 | 413 | 878.3 | 0.7 | 0.65 |
| | DIV × T | 4 | 148.0 | 419 | 882.5 | 37.0 | <0.0001*** |
| | CD × T | 6 | 31.0 | 407 | 847.2 | 5.2 | <0.0001*** |
| | DIV × CD × T | 12 | 14.4 | 395 | 832.8 | 1.2 | 0.27 |

**Note:**
The analysis used a GLM with binomial error distribution and probit link function (significant codes: $^{***} = P < 0.001$ and $^{*} = P < 0.05$).

intensities showed no clear trend (Figs. 3C–3F); however, on average, they were close to 0.5, showing an equal contribution of AC and BM to germination in the mixtures (Fig. 3I).

## Germination under laboratory conditions

In the laboratory trial 1, the DIV effect was significant at $p < 0.05$, and the SPA effect was almost significant at $p < 0.1$ (Table 2). On average across diversity treatments, the single seed spacing showed higher germination than the double seed spacing by 6.6%. Notably, in the single seed spacing, the germination of AC, BM, and the Mix were higher than the double seed spacing by 17.3%, 6.8% and 1.7% (Fig. 4A). Although AME was not significantly affected by seed spacing, a negative AME was observed at single seed spacing, with germination in the mixture being 6.7% lower than in the average of the monocultures. In contrast, the double seed spacing showed a trend of positive AME, though this was not significantly > 0 (Fig. 4B). The PGR showed that both of the species almost germinated equally in the mixture independent of seed spacing (Fig. 4C).

In the laboratory trial 2, the binomial GLM showed that the main effects DIV, *T*, CD, DIV × *T*, and the interactions DIV × *T* and CD × *T* were significant (Table 2). For both species in monoculture and mixture, the highest germination percentage was observed at 20% of WHC and was significantly higher than 12 °C and 28 °C by 25.1% and 19.9%, respectively (Figs. 5A–5C). On average across drought intensities, the germination of $BM_{mono}$ was significantly higher than $AC_{mono}$ at 12 °C and 20 °C by 27.7% and 21.0%, respectively, but not at 28 °C. In the mixture, the improved germination of both species was temperature dependent. Specifically, at 12 °C, the germination of both species surpassed their monocultures; this effect gradually decreased with increasing temperature until no difference was observed at 28 °C between the monocultures and the mixture (Figs. 5A–5C).

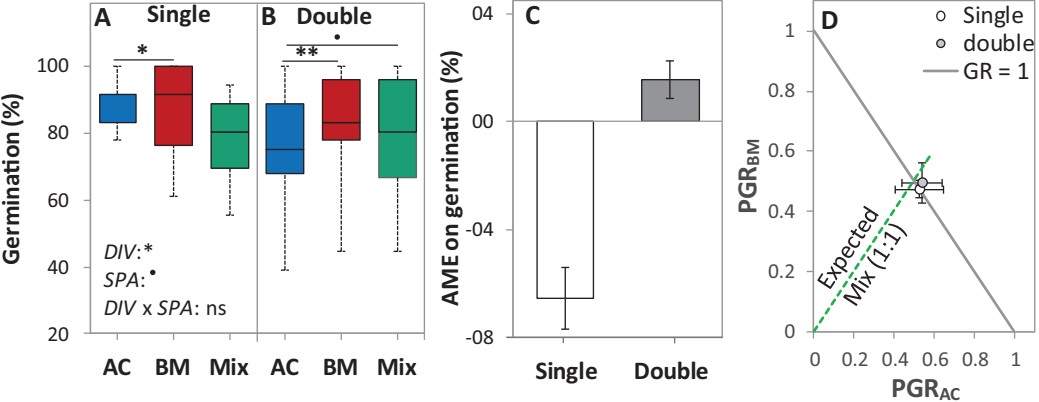

**Figure 4 Germination percentage of alsike clover (AC), black medic (BM), and a 1:1 mixture (Mix) of the two species in the laboratory trial 1.** Germination percentage (%; A and B), absolute mixture effect on germination (AME; C), and partial germination ratio of both species (PGR; D) are calculated for seeds sown (18 seed tray$^{-1}$) in a single (distance between each seed = 1 cm) and double (two seeds in pairs) spacing (SPA). The boxes of the boxplot represent the median, and 25th and 75th percentiles, whereas whiskers represent the maximum and minimum values. Asterisks indicate significant effects based on the analysis of deviance for the generalized linear model of *DIV, SPA* and *DIV × SPA*; $^{**}P < 0.01$, $^{*}P < 0.05$, $^{·}P < 0.1$, and ns = not significant. The absolute mixture effect was calculated by Eq. (1). Positive values indicate positive mixture effects and vertical bars represent standard error of $n = 12$. Partial germination ratio was calculated by Eq. (2); the solid grey lines and the broken green lines as in Fig. 2.

The results of the AME, GR, and PGRs showed specific effects on germination. Specifically, the absolute mixture effect (AME) on germination, that is, the difference between the observed value in the mixture and the average value of the monocultures was significantly affected by temperature but not by water availability. On average of all drought intensities, the AME at 12 °C was significantly positive and higher than the AME at 20 °C and 28 °C by 67.0% and 95.4%, respectively (Fig. 6). Increasing water availability decreased AME at 12 °C until the AME was similar at 100% WHC for the three temperatures.

The GR and the PGR$_{BM}$ were significantly affected by the temperature levels, particularly at 12 °C, GR was > 1 at the different drought intensities, but this was only significant at 25% WHC (Fig. 7A). Notably, there was no consistent directional effect of drought intensity on GR. PGR$_{BM}$ showed a clear trend of being reduced by increasing temperature, namely PGR$_{BM}$ was significant > 0.5 at 12 °C (Figs. 7A–7C).

## DISCUSSION

Our overall results indicate that mixing legume species may significantly affect seed germination, compared to the monocultures of the two species. However, there was no consistently positive or negative direction of the mixture effect. Instead, we observed both significantly positive and negative responses of germination to mixing the two species. Further, the mixture effect was dependent on other factors, notably temperature level as well as water availability. In particular, there was a significant positive mixture effect at sub-optimal temperature suggesting interspecific facilitation at 12 °C. In the following, we first discuss the effects of species identity on germination, as an important prerequisite

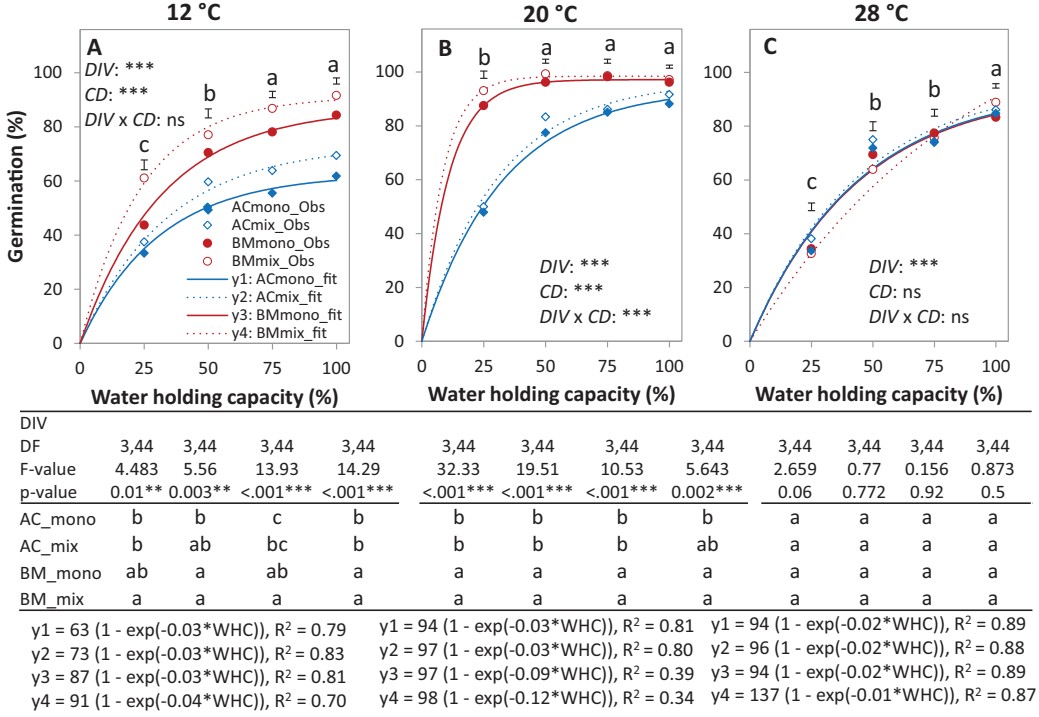

**Figure 5 Germination percentage of alsike clover (AC) and black medic (BM) in monocultures and a 1:1 mixture of the two species in response to four drought intensities and three temperature levels.** The drought intensities (CD) represent 100%, 75%, 50% and 25% of water holding capacity (WHC) and the three temperature levels represent 12 °C (A), 20 °C (B) and 28 °C (C). Symbols represent the observed germination ($n$ = 12) and lines represent the fitted curves by Eq. (3). Vertical bars represent Tukey's HSD ($p < 0.05$) for the differences among the two species in the monocultures (mono) and mixture (mix) at a given WHC (%). Different letters above the vertical bars represent Tukey's HSD ($p < 0.05$) and indicate significant differences among the WHC. Asterisks indicate significant effects of the *DIV, CD* and *DIV × CD* based on the analysis of deviance for the generalized linear model; ***$P < 0.001$, **$P < 0.01$, *$P < 0.05$, ns = not significant.

to understand effects in the mixtures. We then present potential mechanisms that may underlie these observations and discuss our findings with respect to our initial hypothesis.

We assume that any mixture effects on germination percentage would be caused by interactions among seedlings, not among seeds. The existence of a general mechanism explaining the communication among the seeds prior to emergence would be highly intriguing. However, common sense dictates that relatively inert seeds are less likely to interact physically or chemically, in their surroundings, than the physiologically active seedlings, thus may not affect their neighbors. Unfortunately, the differences between seed and seedling effects on germination cannot be explicitly distinguished in the field.

## Effect of species identity on germination

Under controlled conditions, the germination of BM was significantly higher at 12 °C and 20 °C than at 28 °C (Figs. 5A and 5B). This observation is consistent with the fact that the optimal germination for BM is between 10 °C and 20 °C (*Sharpe & Boyd, 2019*) but in contrast to a study stating that the optimum temperature for germination of BM is 26 °C (*Tribouillois et al., 2016*). The sensitivity of BM germination to high temperature is also in

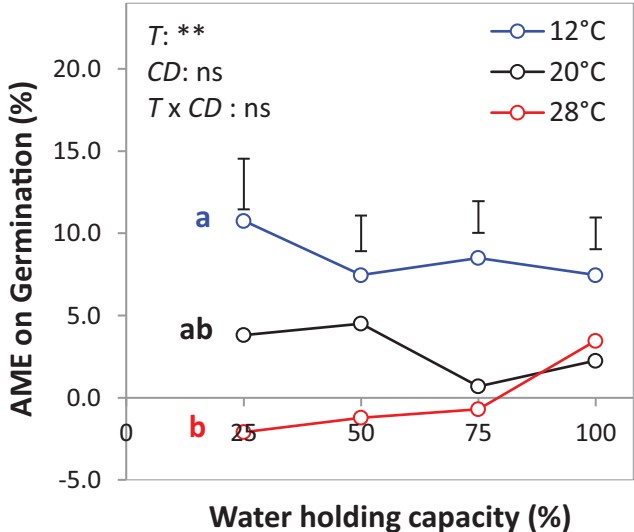

**Figure 6 Absolute mixture effect (AME) on germination (%) of alsike clover (AC) and black medic (BM) in a 1:1 mixture of the two species in response to four drought intensities and three temperature levels.** The drought intensities (CD) represent 100%, 75%, 50% and 25% of water holding capacity (WHC) and the three temperature levels represent 12 °C (A), 20 °C (B) and 28 °C (C). The absolute mixture effect was calculated by Eq. (1); positive values indicate a positive mixture effect. Vertical bars represent Tukey's HSD ($p < 0.05$) among the different temperature levels at a given WHC (%) of $n = 12$. Different letters next to the observed data of the temperature levels indicate significant differences on average of all the drought intensities, based on Tukey's HSD ($p < 0.05$). Asterisks indicate significant effects of the $T, CD$ and $T \times CD$ according to ANOVA; $**P < 0.01$ and ns = not significant.

contrast with a number of studies that have reported BM to be adapted to warm conditions, though this observation may refer to the response after crop establishment or at later growth stages (*Döring et al., 2013*; *Elsalahy et al., 2019*). The ability of AC to germinate at 12 °C and 20 °C was significantly lower than BM (Figs. 5A and 5B). This observation was in contrast to our expectation as it was supposed for AC to germinate better than BM, specifically at 12 °C based on its supposed geographic origin in northern latitudes (*Shaeffer, Wells & Nelson, 2017*). In addition, many studies reported the adaptation of AC to cool conditions (*Döring & Boufartigue, 2013*; *Sheaffer et al., 2003*).

The temperature influenced germination percentage but not germination dynamics of both species where both species show the maximum percentage of germination at 20°C and decreased at temperatures lower and higher than the maximum temperature (Figs. 5A–5C). This germination dynamic has been reported for several species from the family Fabaceae (*Tribouillois et al., 2016*). The inability of AC and BM to reach a high percentage of germination at high temperatures (Fig. 5C) is most likely due to the presence of hard seeds in the seed lot (*Uzun & Aydin, 2004*). This property is well described for *Medicago* and *Trifolium* species, whose hard seed coats may lead to low germination as high temperature deactivates the enzymes required for releasing some volatile compounds that are essential for breaking dormancy (*Motsa et al., 2017*).

The germination percentage of BM was consistent at all levels of water availability at the optimal temperature level (Fig. 5B), in contrast to a study reporting that germination of

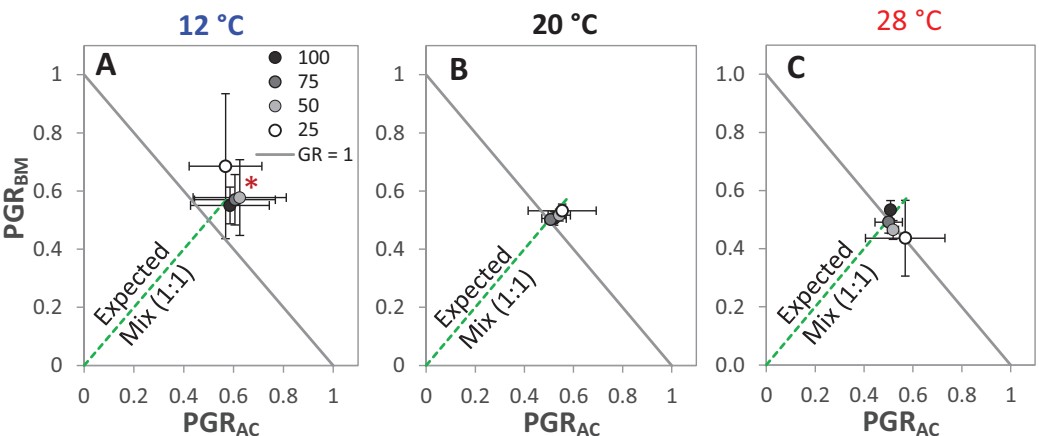

**Figure 7 Partial germination ratio of alsike clover (PGR$_{AC}$) and black medic (PGR$_{BM}$) in a 1:1 mixture of the two species in response to four drought intensities and three temperature levels.** The drought intensities (CD) represent 100%, 75%, 50% and 25% of water holding capacity (WHC) and the three temperature levels represent 12 °C (A), 20 °C (B) and 28 °C (C). The drought intensities are visualized in black gradient color. The solid grey lines correspond to a GR = 1 and the broken green lines correspond to the expected PGR for the mixture. Asterisks above some data points represent a significant increase in GR > 1 ($P < 0.05$) according to Welch's $t$-test; *$P < 0.05$.

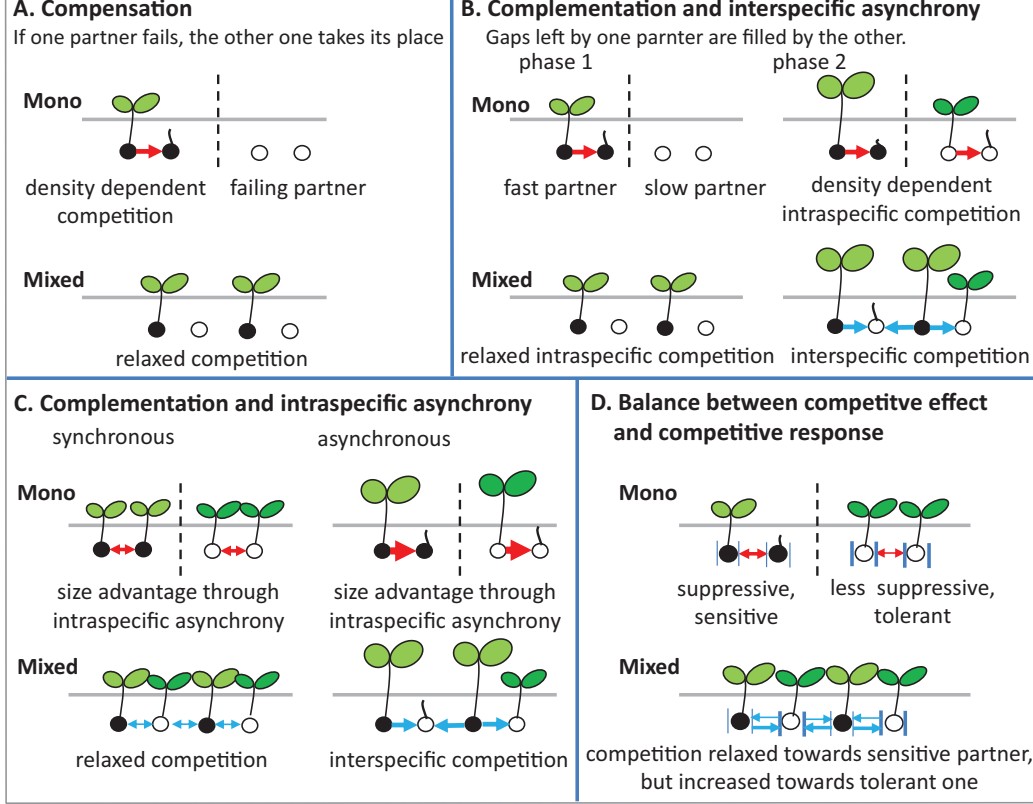

**Figure 8 Schematic illustrations describe mechanisms of mixture effect on germination.** Interspecific competition is represented in blue arrows and intraspecific competition is represented in red arrows; further explanations are in the text of the parts (A–D).

most Fabaceae species, among them BM, are sensitive to low water availability and better suited to rainy climates (*Tribouillois et al., 2016*). The lower germination percentage of AC than of BM under dry conditions (Figs. 2A, 3A–3D, 5A and 5B) is in agreement with some studies reporting AC's sensitivity to dry conditions (*Chapman, Dodds & Keoghan, 1990*). However, in a recent study by *Elsalahy et al. (2020)*, it was found that the drought resilience of AC is higher than of BM. This finding indicates that the response of AC to water availability is phenologically dependent that is, is based on the stages of plant development (*Hatfield & Prueger, 2015*).

In summary, the response of germination to environmental factors (temperature and water availability) differs between the two studied legume species AC and BM, but the complexity of interactions among these abiotic factors makes it difficult to predict which of the species will win the germination race under a given set of environmental circumstances.

## Potential ecological mechanisms: how mixing species may affect germination

There are several potential ecological mechanisms how mixing plant species could affect germination. Several mechanisms have been described for productivity (i.e., plant biomass and grain yield) (*Bedoussac et al., 2015*), and in principle, some of these mechanisms may also be relevant for germination (Fig. 8).

(1) Compensation: The mechanism of compensation means that if one of the partners in a mixture fails, the other one takes its place. It has been shown to be responsible for stabilizing yields of plant mixtures in response to disease (*Creissen, Jorgensen & Brown, 2013*), but can also be underlying gains of mean productivity in mixtures. In the context of germination, we assume that one of the partners in a mixture fails to germinate at all, for example, because of a species-specific pathogen only affecting this partner. If there is intraspecific competition on germination in the non-failing partner, that is, germination is reduced by high density, and then mixing species leads to relaxed competition in the non-failing partner, so that overall germination is higher in the mixture than in the average of the two monocultures (Fig. 8A). Indeed, we did observe reduced germination at higher seed densities, that is, negative density dependence, in the field (Fig. 2A) and in the climate chamber (Fig. 4A), suggesting intraspecific competition. Such density dependence of germination has also been found in other species (*Tielbörger & Prasse, 2009*), though in some cases positive density dependence has also been found (*Leverett, Schieder & Donohue, 2018*). Note that in the context of compensation, the reason for failure can vary, and that in principle it may also locally vary which of the partners in a field is affected by a failure-inducing factor, as long as the spatial zone of competition is considerably smaller than the locally affected area leading the other species to fail. The point is that a locally acting mortality factor only affecting one partner will reduce competition for the remaining partner. A negative mixture effect, however, would be unlikely to emerge from compensation.

(2) Complementation (or complementarity) means that the two partners in a mixture complement each other in their resource use, either spatially or temporally, so that jointly
they can use a greater amount of the resource than if they are in single stands. Gaps left by one component are filled by the other one. Evidence for complementation has been presented for various species mixtures (*Von Felten & Schmid, 2008*; *Hooper, 1998*; *Xiao et al., 2018*). Here we assume that complementation is primarily based on *temporal* differentiation in resource use as there is (probably) less room for *spatial* complementarity in the very early stage of plant development; germinating plants may or may not be overlapping in their zones of influence, but a strong spatial differentiation between species, as observed, for example, for developed root systems or above-ground plant architecture affecting light capture, is, in our view, unlikely in germinating plants.

In this case, this means that there is a fast and slow germinating partner, that is, there is asynchronous behavior of the two species. Evidence for such interspecific asynchrony is an observed increase in the length of the radical of BM in comparison with AC at 12 °C in the laboratory trial, suggesting a faster germination rate of BM than of AC. As our focus, in the laboratory experiment, was on germination responses rather than on the subsequent seedling growth, we recorded our observations for the asynchronous germination by photos (Fig. S2). Our finding regarding asynchronous germination is consistent with the crop growth rate of AC and BM on the base of biomass production, as we have reported in previous studies, that BM is faster growing than AC (*Elsalahy et al., 2020*). In addition, in both field trials, BM was dominant at the early stage of crop establishment.

For temporal complementation mechanisms to come into force, we need to assume, as for compensation, that density dependent intraspecific competition affects germination, so that with higher seed density germination is reduced within both monocultures. As we have argued above, this assumption of density dependent germination rate is supported by our data. In this case, the faster partner has an increased germination rate in the mixture than in its monoculture, because of relaxed competition, since its slower neighbors in the mixture are not yet competing (phase 1 in Fig. 8B).

The outcome of interest of this scenario, that is, the mixture effect, will then depend on the relationship between inter- and intraspecific competition in phase 2 when the slow partner germinates, as follows. First, we assume that interspecific competition (blue arrows) and intraspecific competition (red arrows) are equal in size. In that case, it does not matter for the slower partner whether it is neighbored by its own kind as in the monoculture, or by the faster partner as in the mixture, and therefore germination of the slower partner will not be affected by mixing. Because of the increased germination of the faster partner in the mixture than in the monoculture, however, the overall mixture effect will be positive. It is now easy to see that this positive mixture effect can be reduced, and even turn negative, if the size advantage of the faster species in phase 2 leads the slower partner to experience stronger competition from its neighbors than in its own monoculture (*Connoily & Wayne, 1996*). In this case, germination of the slower partner will be reduced in the mixture in comparison to its monoculture, and the positive mixture effect on the faster species and the negative effect on the slower species will cancel each other out, or, with a larger relative outcompeting effect, even lead to negative mixture effects.

(3) Intraspecific asynchrony: Asynchronous germination, as described above for the two species, can also occur within each of the species, notably at lower than optimal temperature. It is expected that with germination more spread out over time because of low temperature (*Groot, 2020*; *Luo et al., 2018*), monocultures and mixtures will be differently affected, but that this difference again depends on the relationship between intra- and interspecific competition. Here, we assume that interspecific competition is weaker than intraspecific completion, so that this case falls under the heading of complementation. Then, temporally spread-out germination within a species (intraspecific asynchrony) means that—because of the size advantage of a neighbor of the same species—germination losses in the monoculture through competition will be higher than if germination is more synchronous (Fig. 8C). However, in the mixture, this effect of asynchrony may be less marked if interspecific competition is weaker than intraspecific competition. This means that for lower temperatures a higher mixture effect on germination could be predicted, and this is indeed what we found in the laboratory experiment with controlled temperature (Fig. 6). However, in the field where temperature levels were markedly lower in 2017 than in 2016 (Fig. 1B), this expectation was not confirmed as the AME was lower in 2017 than in 2016 (Fig. 2B).

(4) Competitive effect and response: This mechanism refers to the balance between competitive effect (i.e., how suppressive a species is) and competitive response (i.e., how tolerant it is when faced with competition) (*Goldberg & Landa, 1991*). We assume that the two partners in the mixture differ in their competitive traits such that one partner (A) is more suppressive (high competitive effect on a target plant), but also more sensitive (low tolerance), while the other one (B) is less suppressive, with milder competition towards a target plant, but less sensitive, that is, exhibiting a more tolerant competitive response (Fig. 8D). Here it can be seen that, as before, the mixture relaxes the competition towards one of the partners, in this case, the more sensitive one, because the immediate neighbor is less suppressive than in the monoculture of the suppressive partner. However, in the mixture, this less suppressive partner is also neighbored by a more suppressive partner than in its monoculture, which could mean that germination of the less suppressive partner is reduced. This case will depend on the relationship between competitive effect and competitive response. If the high tolerance of partner B matches the suppressive competitive effect of partner A, then germination will not be reduced in B. If, on the other hand, the suppressive force is stronger, there will be a reduction of B's germination and a concurrent reduction in the mixture effect.

These considerations also lead to the question of how mixture effects on germination might be influenced by the presence of weeds in the field. Here, the prediction is that the mixture effect will be reduced, that is, diluted, by the presence of (suppressive) weeds. The likelihood that the more sensitive crop partner is neighbored by weed species with high competitive suppressiveness is the same in the monoculture as in the mixture; however, the relaxation of competition through the absence of a same-species neighbor with high competitive ability will be lost through the weeds' presence. However, in the field, our data was not consistently in line with this prediction, as we observed no clear relationship between weed density and AME (Fig. 2B). Further, completely removing

weeds as in the pot trial and in the climate chamber trials, did not consistently lead to higher AME (e.g., Figs. 3H, 4B and 6) than in the field.

(5) Facilitation: A last, currently more speculative, mechanism of interaction between the two species is the production of VOCs that may promote (or reduce) the germination of one or both species (*Motsa et al., 2017*). While it is easy to argue that the production of VOCs, their accumulation in the growing medium, or their decomposition rate may have been affected by the varied environmental conditions to explain the (chaotic) dependency of the mixture effect on the abiotic factors, it is impossible to conclude from our data whether, or in which ways, VOCs may have been involved in the mixture effects.

## CONCLUSIONS

As our results show, there was no consistent mixture effect on germination. However, we did find significant effects of mixing on germination, both positive and negative, under some experimental conditions.

It is not possible to infer causality from our experiments in relation to the observed effects of mixing species on germination. However, as we have argued in the discussion, there are several potential mechanisms that may, when combined, explain both directions of mixture effects on germination. At the same time, the ecological interactions that may be involved in these mechanisms are complex and sometimes even difficult to measure at the spatial scale that is relevant for germinating seeds. This means that it may be almost impossible to predict, under realistic field conditions, the direction and size of mixture effects on germinating (crop) plants. This may be different when more dissimilar plant species are combined, or when severe stress factors only affect one of the partners in the mixture. Future research dealing with mixing effects on germination should, therefore, focus on these conditions. In this case, we suggest that it will be increasingly important to understand the compensation ability of the partner species in monocultures and mixtures. Our research has highlighted that mixture effects may depend on environmental conditions, though not in easily predictable ways. If and insofar as this general observation is also valid for mixture effects on yields, it will be necessary to re-think strategies for optimizing crop mixtures for use on individual farms. In this case, it will be necessary to create resources for farmers to help them to adopt and adapt this practice to their own farm context.

## ACKNOWLEDGEMENTS

We thank the Nicolai Schmidt, Gero Carus, Cansu Cali, Manuel Müller, Anne Donath and the staff at the Humboldt-University, department of Agronomy and Crop Science for support during the experiments.

### Funding

This work was funded by the Yousef Jameel Scholarship at Humboldt University, Berlin, Germany, and the Ekhaga Foundation as a part of Ph.D. Project. The funders had no role

in study design, data collection and analysis, decision to publish, or preparation of the manuscript.

### Grant Disclosures
The following grant information was disclosed by the authors:
Yousef Jameel Scholarship at Humboldt University, Berlin, Germany, and the Ekhaga Foundation as a part of Ph.D. Project.

### Competing Interests
The authors declare that they have no competing interests.

### Author Contributions
- Heba Elsalahy conceived and designed the experiments, performed the experiments, analyzed the data, prepared figures and/or tables, authored or reviewed drafts of the paper, and approved the final draft.
- Sonoko Bellingrath-Kimura performed the experiments, authored or reviewed drafts of the paper, and approved the final draft.
- Timo Kautz analyzed the data, authored or reviewed drafts of the paper, and approved the final draft.
- Thomas Döring conceived and designed the experiments, performed the experiments, analyzed the data, prepared figures and/or tables, authored or reviewed drafts of the paper, and approved the final draft.

### Data Availability
Raw data are available in the Supplemental Files.

### Supplemental Information
Supplemental information for this article can be found online at http://dx.doi.org/10.7717/peerj.10615#supplemental-information.

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
