# Peer review of "Effects of mixing two legume species at seedling stage under different environmental conditions"

_PeerJ, doi:10.7717/peerj.10615_

## Round 0.1 · original submission · Major Revisions

Dear Dr. Elsalahy,

Please find below the comments of two independent reviewers. Based on that, I agree that, besides the interest of the paper, it deserves major changes before being accepted form publication. Thus, I invite you to address all questions raised by the reviewers, paying particular attention to:
- the experimental design and statistical analysis
- the improvement of the structure of the discussion with updated references

Looking forward to receive the modified version,
Sincerely

Ana I. Ribeiro-Barros

Reviewer 1 ·

Basic reporting

The manuscript contain new and significant information to justify publication. It is a study with overall adequate presentation of experimental results,
It is a complementary experiment study useful in agriculture and in research academic. The concept of the study is well investigated and approved by different trials in lab, pot and field.
Sufficient field backgroung/context provided.
Some additions are needed: this paper has potential and is interesting in productivity for adaptation to climate change, however in the current form needs to be reviewed.
Concerning title: I suggested to the authors to adjust the title. I propose (potential use of association (mixture) legumes species as a sustainable technique at seedling stage under the different environmental condition for the intercropping system)
The abstract clearly and accurately describes the content of the article.

Experimental design

The experimental and/or theoretical methods described comprehensively.
Methods described with sufficient information.
The statistical procedure that you used is the most correct for what you are analyzing.

Validity of the findings

The statistical procedure that you used is the most correct for what you are analyzing.
Conceptual of manuscript could be more original if author pay more attention particularly in the conclusions.
Because the study is agronomic investigation give highlights at farmer level the agronomic potential of association pasture legume introductions on droughty soils.

Additional comments

I invite the authors to submit a major revision of the manuscript before a final decision is reached

Annotated reviews are not available for download in order to protect the identity of reviewers who chose to remain anonymous.

·

Basic reporting

The title of the article is misleading, only two legumes were used. Proofreading is required. In the introduction I am missing current articles on this topic: DOI: 10.1111 / grs.12257; 10.1002 / agj2.20045; 10.1111 / jac.12337; 10.2134 / agronj2017.12.0753; 10.1139 / cjps-2017-0183 etc.

Experimental design

Methods, resp. in particular, the design of the experiment is completely unclear. Proper and clearly repeating field experiments would suffice. According to the results and discussion, the others are completely irrelevant to the topic and hypotheses of the thesis.
Why was 3 times in 2016 and 4 times in 2017. Cleary explain pot (samples), 9 variants? 8 section and 3 blocks - how area?. You used just only two legumes and their mixture... Missing table and design of experiment.

In my opinion, laboratory and pot experiments are not suitable for confirming or refuting hypotheses. They cannot simulate abiotic and biotic stresses in all directions.

Statistical models, resp. evaluation is based only on coefficients? Have they been successfully verified somewhere before? Published?

Validity of the findings

The validity of the results is OK. But in general they are very unconvincing given the hypotheses. But even that is the result.

I consider this chapter (Potential ecological mechanisms: How mixing species may affect germination) a literary review. Most results are not significant or have a large variance. Causality cannot be inferred from them.

Additional comments

The article should be clarified incl. methods. Comment only on clearly conclusive results and draw clearly verifiable conclusions from them.

---

## Round 0.2 · Minor Revisions

Dear Dr. Elsalahy,

Thank you for sending the revised version of your manuscript - Effects of mixing two legume species at the seedling stage under different environmental conditions. Although this version is almost acceptable for publication, I suggest that you perform the minor revisions suggested by reviewer 1.

Sincerely,
Ana I. Ribeiro-Barros

Reviewer 1 ·

Basic reporting

I thank editors for the confidence given to me to review the manuscript on “Effects of mixing legume species on germination” (Manuscript ID: #50696)
I think authors for patience and time are given for improve the manuscript. The manuscript is revised carefully.
Comments and suggestions of reviewers were taken into account and have included all the reviewers’ comments responded to them individually in a point-by-point response, in preparing the revised version of the manuscript.
Changes made in the manuscript and responses of authors are satisfactory and that the newly revised manuscript is acceptable for publication.
The two versions of the revised manuscript, with track changes and with all track changes accepted are revised and updated. The required additions have been added to the manuscript.
The title was adjusted as suggested and improved. Comments and remarks through the manuscripts taken into account in the revised manuscript.
The abstract clearly and the experimental and/or theoretical methods described comprehensively and improved in the revised version and supplemented file for explanations added.
Sentences needed to be deleted: Line 50- Line 52: "Cover crops are frequently sown during March-April in temperate climates and October-November in subtropical areas according to the fallow-period duration determined by the cash-crop succession"
Field FU9: please define composition to be clear to the reader, is it ready substrate, what is composition? Precise for each texture sandy clay loam or sandy loam or….
References are well checked only there is a minor revision in yellow color comments in added manuscript version revised. Please check the version uploaded for a minor revision. All points are already made in the revised manuscript.
A graphical abstract prepared demonstrates the high probability of increasing the germination percentage in the case of mixing the two legume species that can be added to supplementary data for publication.
I advise as perspectives to identify and analyze the crop mixtures grown in farmland (with the different farmers) with a view to creating resources to help farmers to adopt and adapt this practice to their own context.
The manuscript contains new and significant information (even research needed to be improved in the field in farmland in the future) to justify publication. The study with an overall adequate presentation of experimental results discussion; it is a complementary experiment study useful in agriculture and in research academic. I suggest for publication after taking into account minor revision.

Experimental design

The study with an overall adequate presentation of experimental results discussion; it is a complementary experiment study useful in agriculture and in research academic.
Into revised manuscript author improved this part with one figure

Validity of the findings

Authors improved all parts of the manuscript as suggested by reviewers
the benefit to the reader is clearly stated.
Conclusions are well stated and improved

Additional comments

I advise as perspectives to identify and analyze the crop mixtures grown in farmland (with the different farmers) with a view to creating resources to help farmers to adopt and adapt this practice to their own context.

Annotated reviews are not available for download in order to protect the identity of reviewers who chose to remain anonymous.

·

Basic reporting

The authors improved the manuscript very conscientiously and carefully.

Experimental design

After the revision, the methods are clearly explained incl. scheme of experiment in the S

Validity of the findings

Now is clear and better explained.

Additional comments

Thank you for revised version. It is clear that a thorough revision of the manuscript has helped significantly.

---

## Round 0.3 · accepted · Accept

Dear Dr. Elsalahy,

Regarding your manuscript - Effects of mixing two legume species at seedling stage under different environmental conditions - it is my pleasure to inform you that it is now accepted for publication. Thank you for carefully addressing the peer-reviewers' suggestions and keeping the quality standards of PeerJ.
Sincerely,
Ana I. Ribeiro-Barros